# Fit-for-purpose quantitative liquid biopsy based droplet digital PCR assay development for detection of programmed cell death ligand-1 (PD-L1) RNA expression in PAXgene blood samples

Dennis O'Rourke[1‡], Danyi Wang[1‡], Jorge F. Sanchez-Garcia[1], Maria Perella Cusano[1], Waldemar Miller[2], Ti Cai[1], Juergen Scheuenpflug[3], Zheng Feng[1]*

1 Global Clinical Biomarkers & Companion Diagnostics, Translational Medicine, Global Development, EMD Serono Research and Development Institute, Inc. a division of Merck KGaA, Billerica, MA, United States of America, 2 Biosample Informatics and Biobanking, Clinical Trial Management, Global Clinical Operations, Global Development, Merck KGaA, Darmstadt, Germany, 3 Global Clinical Biomarkers & Companion Diagnostics, Translational Medicine, Global Development, Merck KGaA, Darmstadt, Germany

‡ These authors share first authorship on this work and contributed equally to this work.
* zheng.feng@emdserono.com

## Abstract

Development of a clinically applicable liquid biopsy-based test for PD-L1 mRNA expression would be beneficial in providing complementary evidence to current immunohistochemistry assays. Hence, we report the development of a fit-for-purpose assay for detection of blood PD-L1 mRNA expression using droplet digital polymerase chain reaction (ddPCR). Taq-Man® assays were selected based on coverage of the PD-L1 gene and were tested for linearity and efficiency using real-time quantitative PCR. Four reference genes were analyzed in positive control cell line (A549 treated with interferon gamma, [IFN γ]) genomic DNA. The PD-L1 primer/probe sets were also evaluated in ddPCR for limit of blank, limit of detection, and precision. Finally, thirty-five healthy volunteer samples were evaluated to establish a baseline level of PD-L1 expression. In ddPCR, the limit of blank was determined to be 0 copies and the limit of detection was determined to be less than or equal to 19 copies of PD-L1. The average intra-run coefficient of variation in the ddPCR assay was 7.44% and average inter-run CV was 7.70%. Treatment of A549 cells with IFN γ resulted in a 6.7-fold increase in PD-L1 expression (21,580 copies in untreated cDNA versus 145,000 copies in treated cDNA). Analysis of healthy human samples yielded a median value of 1659 PD-L1 copies/µL with a range of 768–7510 copies/µL. The assay was transferred to an external service provider and results from our in-house experiments and those conducted externally shows a correlation of 0.994. In conclusion, a fit-for-purpose liquid biopsy-based, purely quantitative ddPCR assay for the detection of PD-L1 mRNA expression was developed and validated using PAXgene RNA blood samples. Linearity, reproducibility, limit of blank and limit of detection were measured and deemed suitable for clinical application. This ultra-sensitive

**Data Availability Statement:** All relevant data are within the manuscript and its Supporting Information files.

**Funding:** This study was funded by EMD Serono Inc., a part of Merck KGaA Darmstadt, Germany. EMD Serono/ Merck KGaA is the employer of all authors (with the exception of Maria Cusano, who was previously employed by EMD Serono during the time the experiments were conducted) at the time of submission. Merck KGaA/ EMD Serono is a commercial company who provided support in the form of salaries for authors DO, DW, JSG, MC, WM, TC, JS, and ZF, but did not have any additional role in the study design, data collection and analysis, decision to publish, or preparation of the manuscript. The specific roles of these authors are articulated in the 'author contributions' section.

**Competing interests:** The authors declare that they have no competing interests which may influence this publication or alter our adherence to PLOS ONE policies on sharing data and materials.

liquid biopsy ddPCR assay has promising clinical potential in screening, longitudinal monitoring and disease progression detection.

# 1. Background

Programmed cell death ligand-1 (PD-L1) has been identified as an efficacious target to develop immune checkpoint inhibitors and has drastically evolved the treatment paradigm for advanced cancers.

PD-L1 expression detected by immunohistochemistry (IHC) in tumor tissue has been used as a predictive biomarker for therapeutic response to PD-1/PD-L1 blockade treatment in various cancers such as non-small cell lung cancer (NSCLC), [1,2]. However, given the dynamic interplay between the PD-1/PD-L1 pathway and the tumor microenvironment, as well as the heterogeneity of immune milieu among different tumor types (hot tumor vs. cold tumor), it has been recognized that tissue (especially core needle) biopsy may not accurately reflect the entirety of the tumor microenvironment. When selecting PD-L1 positive subjects, multiple core biopsy has demonstrated a higher sensitivity than a single core biopsy [3].

Furthermore, many factors affect the tumor immune environment, which determines the equilibrium of immune surveillance and degree of tolerance and adds to the complexity of assessing tumors based on single biopsy samples. Expression of PD-L1 varies during cancer evolution and treatment, which further confounds the characterization of the immune microenvironment landscape. Collectively, invasive tissue biopsies provide limited information to reflect the tumor heterogeneity, and it is not clinically feasible to collect multiple tumor biopsies for longitudinal monitoring of therapeutic efficacy and disease progression.

Therefore, developing a blood derived liquid biopsy-based assay for the detection of circulating PD-L1 mRNA would be beneficial in monitoring disease progression and providing complementary evidence to current IHC assays in both clinical translational research and clinical settings. Such an assay may also provide additional insight to further understand the disease biology and mechanism of action.

Droplet digital PCR (ddPCR) is a purely quantitative technology that follows a comparable workflow to reverse transcription polymerase chain reaction (RT-PCR), which can utilize most standard TaqMan® probe-based assays and has potential for clinical applications. ddPCR partitions each reaction into 20,000 droplets, followed by PCR amplification of template molecules within each droplet. Data is then acquired and offers direct and pure quantification of the gene target of interest without the use of standard curves. The partitioning of each ddPCR reaction and ddPCR technology offers several advantages compared to qPCR, including an absolute quantitative result, improved precision, reproducibility, accuracy, sensitivity and greater tolerance to PCR inhibitors. Nucleic acid quantitation is independent of reaction efficacy, providing greater ability to detect and accurately quantify low-abundance targets [4].

In this study, a fit-for-purpose liquid biopsy assay was developed using PAXgene RNA tubes for the detection of PD-L1 mRNA expression in blood using ddPCR.

# 2. Results

## 2.1 Reference gene selection using qPCR

qPCR was utilized for initial feasibility and assay development efforts as it is a simple and cost-efficient method. ddPCR was run along with qPCR to determine accuracy of qPCR results with absolute quantification.

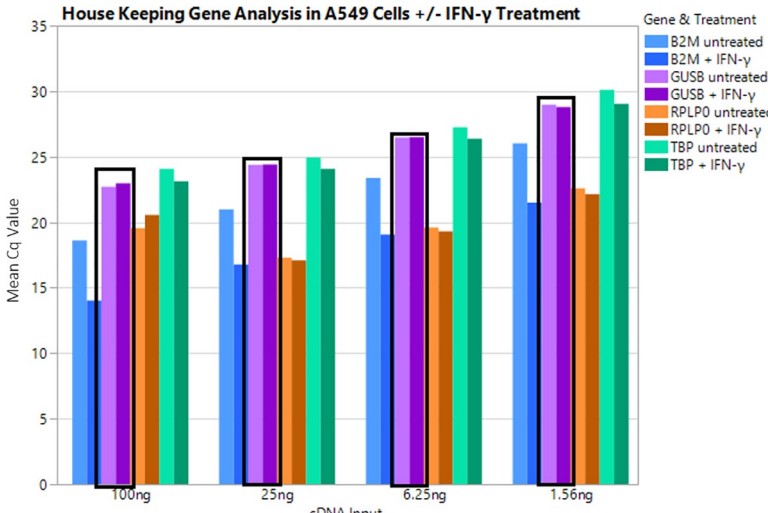

**Fig 1. IFN γ treatment effects on reference gene expression.** cDNA from either A549 untreated cells or A549 + IFN γ treated cells were titrated over four input points from 100ng to 1.56ng and tested in qPCR with four reference genes, B2M, GUSB, TBP, and RPLP0. B2M showed significant difference between untreated and treated samples whereas GUSB, TBP, and RPLP0 showed very little difference between the two samples. GUSB was ultimately chosen as the optimal reference gene and is shown outlined in the black boxes.

Reference gene expression in IFN γ treated and untreated A549 cell lines was compared across three primer/probe kits to determine whether IFN γ treatment influences reference gene expression. Four reference genes Glucuronidase β, Ribosomal Protein Lateral Stalk Subunit P0, β-2 microglobulin, and TATA Box Binding Protein, (GUSB, RPLPO, B2M, and TPB, respectively) were titrated over four cDNA input amounts ranging from 100 ng to 1.56 ng (Fig 1). Raw Cq values are displayed in S1 Table.

Clearly, B2M showed a significantly reduced Cq value upon treatment with IFN γ (Cq value of 18 in untreated sample versus 14 in treated at 100ng input), giving evidence that B2M would not be a suitable reference gene in this assay. TBP shows a slight difference in expression, while GUSB and RPLP0 showed almost no difference in expression, with similar effects when analyzed in healthy volunteer PAXgene blood samples. RPLP0 displayed Cq values much lower than we anticipated in human samples and therefore, GUSB was selected and utilized as a reference gene in future assays.

## 2.2 Linearity of PD-L1 TaqMan⃝R assays

Due to the upper limit of quantitation in ddPCR, the first titration point of 1 x $10^6$ copies was above the dynamic range of ddPCR and excluded. Furthermore, the lowest titration point of 1 copy was below the lower limit of detection in ddPCR and corresponded to a Cq value of approximately 37–39 in qPCR, and, consequently, was also excluded. Therefore, the final standard curve used was a five-point titration ranging from 100,000 copies to 10 copies using a 10-fold dilution factor. All three PD-L1 assays showed very similar PCR efficiency and $R^2$ values (Fig 2 and S2 Table). There was no significant difference in linearity or efficiency among the three primer/probe kits.

The slopes were compared using the compare slopes function in JMP software to model all three equations together in one model and analyze the covariance at each point. Eventually, JMPs 'Compare Slopes' report yielded no significant difference of the individual assays but did show slightly different upper and lower limits for each assay (S1 Fig, S3 Table).

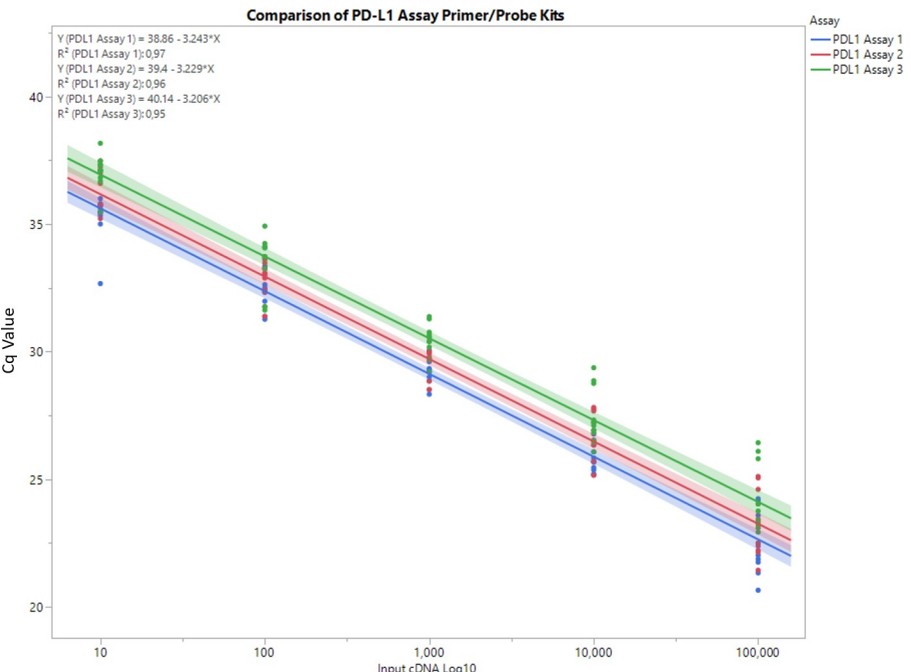

**Fig 2. PD-L1 Assay 1, PD-L1 Assay 2 and PD-L1 Assay 3 standard curves.** Dilutions of the PD-L1 cDNA construct were run on PCR and checked in ddPCR to confirm copy numbers for accurate extrapolation of results. The lines indicate the regression line that is obtained from individual models. Furthermore, a confidence band of the regression line was added (95% pointwise confidence intervals of the mean response). On the top left side of this plot, the formula of the regression line, with X = log10(Copy Number), and the coefficient of determination ($R^2$) are displayed. In our given model, we can utilize $R^2$ as a measurement of linearity, as it can be interpreted as fraction of variability that may be described by a straight line. Therefore, a value close to 1 (or 100%) indicates a good linear fit.

## 2.3 Relative quantitation

Relative quantitation was determined and compared across three primer/probe kits for qPCR using five healthy PAXgene blood samples (Fig 3 and S4A Table) and A549 + IFN γ treated cDNA (not shown) relative to the A549 untreated cDNA with GUSB as a reference gene. The purpose of these experiments was to compare the dynamic range and relative expression of PD-L1 in normal blood and our positive and negative cell line when using qPCR. PD-L1 Assay 3 showed the highest average expression of PD-L1 (0.60) in the tested PAXgene samples whereas PD-L1 Assay 1 (0.56) and PD-L1 Assay 2 (0.43) were lower. However, the median value in the three assays were very similar (0.52, 0.45, and 0.55 respectively). In addition, PD-L1 Assay 2 showed the lowest standard deviation of relative expression (0.09) compared to PD-L1 Assay 1 (0.21) and PD-L1 Assay 3 (0.23).

## 2.4 Absolute quantitation

Absolute quantification was performed to compare the results of both ddPCR and qPCR. In qPCR, a standard curve was created using the PD-L1 construct ranging from 100,000 copies to 10 copies. This standard curve was qualified in ddPCR and then used to extrapolate the concentration of unknown samples in qPCR (S5A Table).

Five healthy PAXgene Blood samples were tested in ddPCR and qPCR using each PD-L1 Assay. PD-L1 Assay 1 showed an average expression of1419.64 copies/μL in qPCR and 3019 copies/μL ddPCR, PD-L1 Assay 2 yielded an average expression of 1429.27 copies/μL in qPCR and 2834 copies/μL in ddPCR and PD-L1 Assay 3 yielded 1723.06 copies/μL in qPCR 2022

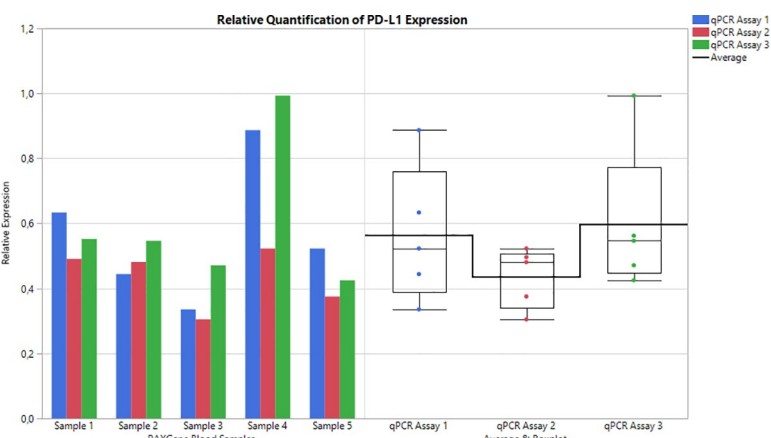

**Fig 3. The relative expression of PD-L1 in 5 healthy volunteer samples normalized to A549 untreated (control) cell line as measured in qPCR (left) and the distribution of relative expression from three PD-L1 assays (right).** The box plot (right) shows the mean expression in the horizontal bold line. The median relative expression is rather consistent between the three PD-L1 assays however the mean expression level shows more variation. Finally, PD-L1 assay 2 shows the tightest distribution of expression among the five healthy samples.

copies/μL in ddPCR (Fig 4, S5A and S5B Table). These differences could be due, at least in part, to the availability of the binding sites required for the primers in each assay to amplify the target DNA. In short, qPCR absolute quantitation yielded an average of 1,523 copies/μL of PD-L1 while ddPCR yielded an average of 2,625 copies/μL of PD-L1 in PAXgene blood samples.

## 2.5 Reproducibility

To better assess the reproducibility of both ddPCR and qPCR for use in detection of PD-L1 in PAXgene blood, we evaluated reproducibility by both inter-assay and intra-assay variation. qPCR Relative expression was defined as expression relative to A549 untreated sample normalized to the GUSB gene whereas qPCR Absolute is derived from a standard curve.

Inter-assay and intra reproducibility was determined by running qPCR and ddPCR in quadruplicate on three separate instances using each of the three selected off-the-shelf primer/

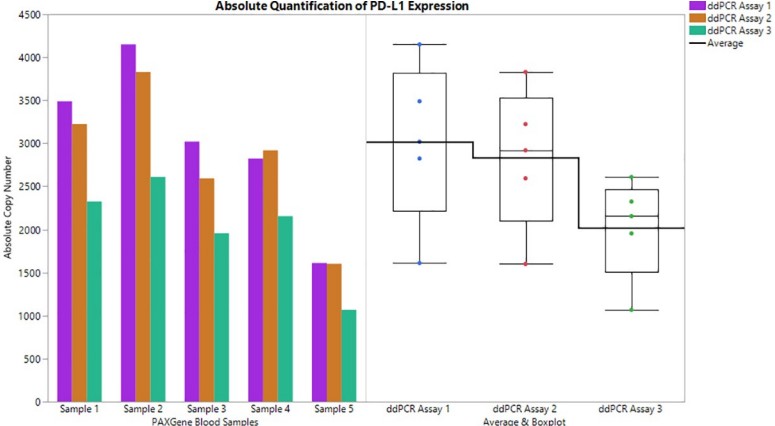

**Fig 4. The absolute quantification of PD-L1 expression in 5 healthy volunteer samples as quantified in ddPCR (left) and the distribution of the absolute expression (right).** The box plot (right) shows the mean expression in the horizontal bold line. In this analysis PD-L1 assay 3 showed lower overall expression than Assay 1 or Assay 2.

**Table 1. Average inter assay CVs (combined average of all PD-L1 assays).**

| Sample type | ddPCR (CV%) | qPCR Absolute (CV%) | qPCR Relative (CV%) |
|---|---|---|---|
| A549 untreated | 11.12 | 25.88 | 0.93 |
| A549 untreated 1:10 | 7.28 | 17.09 | 14.93 |
| A549 + IFN γ | 10.33 | 24.82 | 17.37 |
| Average | 9.57 | 22.6 | 11.06 |

probe kits. Inter-assay CV was calculated based on the average expression (relative or absolute) of each run (Table 1) and Intra-assay CV was calculated for each run and averaged over three separate runs (Table 2).

On average and across samples tested, qPCR had CV values twice that of ddPCR. Absolute quantification with qPCR showed the highest inter-assay CV (average 22.6%) whereas ddPCR (9.57%) and relative quantification with qPCR (11.06%) were comparable (Table 1). Intra assay CVs for individual runs are available (S6A Table).

Similarly, ddPCR showed the lowest intra-assay CV (8.71%) while the qPCR absolute quantification (14.33%) and qPCR relative quantification (14.36%) were very comparable (Table 2). Inter Assay CVs for individual runs are available (S6B Table).

Taking this information together, it was concluded that ddPCR provides more accurate, reliable results for quantification of PD-L1 RNA in PAXgene blood.

In addition, CV data was useful in determining the optimal assay to move forward with in ddPCR. PD-L1 Assay 2 yielded the lowest average inter-assay CV of 5.7% and the lowest intra-assay CV of 6.55% among the three primer/probe sets and thus was selected as the assay to move forward (S6A and S6B Table).

## 2.6 Assay validation with normal donor samples at external vendor

Following in-house method development, eight blinded cDNA samples were delivered to an external vendor for clinical grade assay transfer testing under CLIA/CAP compliance to further evaluate the reproducibility of our results. PD-L1 Assay 2 was used in the assay transfer experiments based on the lowest CV values from in-house experiments. In their experiments, ddPCR was used to quantify PD-L1 and GUSB expression in cDNA from A549 cells untreated and treated with INF γ, and titrations of both sample types. PAXgene blood from two healthy volunteers were also tested. Quantification of PD-L1 expression by ddPCR, as performed by the external service provider, met pre-defined criteria to establish a successful assay transfer. Test samples run at both EMD Serono R&D and the external service provider had a correlation of 0.994 and PD-L1 copy numbers agreed within +/-10% between both sites (Fig 5, S7 Table).

Subsequent validation including limit of the blank (LOB), limit of detection (LOD), and additional reproducibility testing in ddPCR using PD-L1 Assay 2 was performed on an additional 33 normal healthy samples (S8 Table). The LOB, which addresses the experimental background of the assay, was 0 copies for PD-L1 and 11.2 copies for GUSB. The LOD, at

**Table 2. Average intra assay CV average (combined average of all PD-L1 assays).**

| Sample type | ddPCR (CV%) | qPCR Absolute (CV%) | qPCR Relative (CV%) |
|---|---|---|---|
| A549 untreated | 8.15 | 17.11 | 17.93 |
| A549 untreated 1:10 | 10.72 | 8.06 | 8.50 |
| A549 + IFN γ | 7.27 | 16.70 | 16.65 |
| Average | 8.71 | 14.33 | 14.36 |

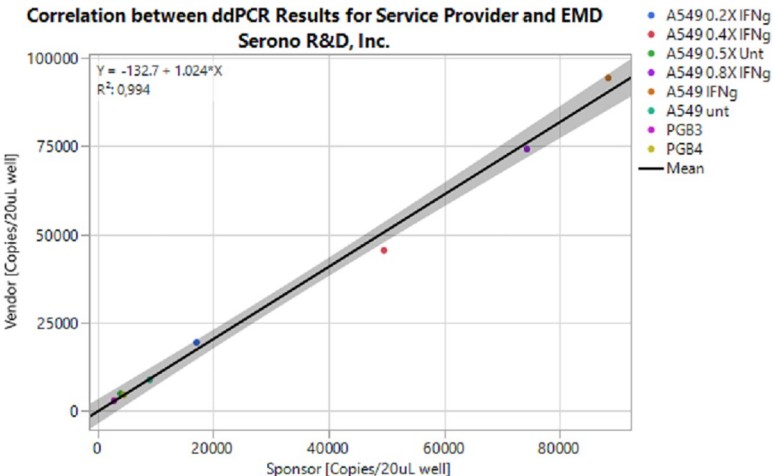

**Fig 5. The correlation between EMD Serono (sponsor) and an external service provider's (vendor) results when performing ddPCR on eight sponsor provider blinded samples was determined and visualized as shown below.** Samples tested include two commercially procured, healthy volunteer samples and A549 cell lines treated and untreated with IFN-γ. Treated cell line was tested neat, at 0.2x, 0.4x, and 0.8x dilution. Untreated sample was tested neat and at a 0.5x dilution. Data points for all samples are shown. A correlation coefficient of 0.994 was calculated.

which target detection is consistent, is 19 copies for PD-L1 and 20 copies for GUSB. Although 19 copies was the average number of PD-L1 copies in the lowest sample dilution analyzed, it is possible that the actual limit of detection is lower if a lower dilution point was run. All intra-run CV values obtained for normal donor samples were ≤20% for PD-L1 and GUSB, which meets the acceptance criteria for intra-run precision [5]. Most samples analyzed (30/32) met the acceptance criteria inter-run precision with CVs ≤ 20% for PD-L1 while 29/32 met the criteria for inter-run precision with GUSB. Furthermore, inter-operator variation was measured at the external service provider and showed high correlation ($R^2$ >0.99) between the operators demonstrating again that our assay is highly reproducible (Fig 6).

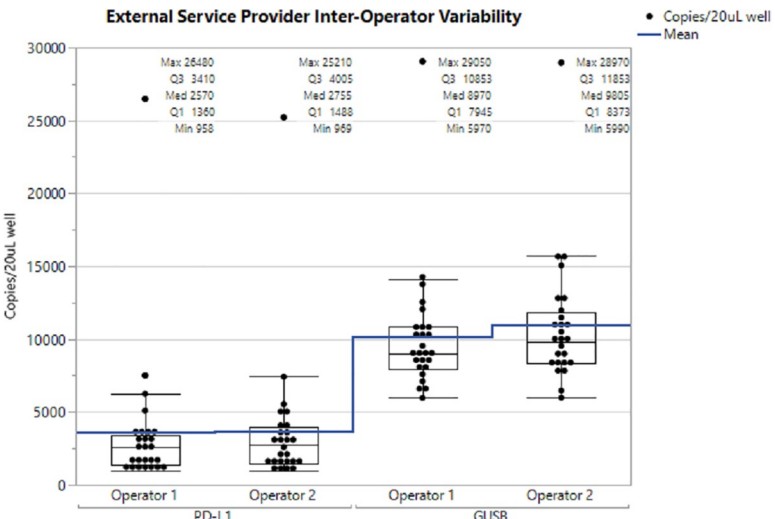

**Fig 6. Comparison of PD-L1 and GUSB copy numbers reported by two operators at an external service provider.** The continuous horizontal line shows the mean copy number and the median is shown in the box plot. As shown in the box and whiskers plot, the assay is highly replicable and concordant between the two operators. One sample produced very high PD-L1 copy numbers and also very high GUSB copy numbers.

## 3. Discussion

In recent years, there have been increasing clinical development efforts using liquid biopsy matrices such as plasma, serum, whole blood, and urine for implementation of biomarker driven clinical translational studies and clinical applications. Liquid biopsies hold potential for patient screening, selection, disease progression monitoring, and relapse monitoring by providing a heterogenous view of the tumor and tumor microenvironment which can possibly reflect tumor metastasis. Therefore, there is an enormous potential in liquid biopsy-based testing to complement current clinical assays.

We developed a purely quantitative ddPCR liquid biopsy-based assay for the detection of PD-L1 using PAXgene RNA tubes. The "fit for purpose" development efforts included early development with synthetic constructs and validation using commercial PAXgene blood samples from healthy controls. This testing included assay performance parameters such as linearity, LOB (0 copies for PD-L1), LOD (<19 copies), variability (CV < 10%) and critical reagent selection, such as reference gene selection and identification and selection of positive cell line material.

We examined three commercially available TaqMan® PD-L1 Assays in both qPCR and ddPCR to determine an optimal primer/probe set and assay for detection of PD-L1 in PAXgene blood samples. The three TaqMan® assays all performed very well with PCR efficiencies from 102%-104% and $R^2$ values >0.995. While the three different PD-L1 TaqMan® assays performed very well, ultimately PD-L1 Assay 2 was chosen for assay validation with the external service provider based on the lowest inter-assay CV and intra-assay CV in ddPCR. In ddPCR, Assay 2 showed similar median and mean expression levels to Assay 1 but with slightly lower range of expression whereas Assay 3 had lower median and mean values.

While our experiments examined the TaqMan® PD-L1 assays in healthy control samples and cell line material for assay performance, comparison of the assays in cancer samples may prove useful. Differences observed between the assays should be thoroughly examined and comparison to tissue-based PD-L1 measurements may prove to be valuable additional information to further characterize the assays.

Following initial experiments, the performance of the qPCR assay was found to be inadequate, prompting the switch to ddPCR which offers absolute quantification, lower limit of detection, and lower inter and intra-assay CV as observed during our experiments. Furthermore, it was determined that the reference gene normalization step required in the qPCR assay induced discrepancy in the final results which negatively impacted the accuracy and reproducibility of the assay. In addition, we found that ddPCR, as a purely quantitative assay, is more sensitive and can more precisely detect PD-L1 expression in blood. Hence, the decision was made to select ddPCR as the platform for developing the PD-L1 blood-based assay. Recently, the first ddPCR IVD kit was approved by the FDA, paving the way for future ddPCR assays to transition into use the clinic [6].

This study also highlighted the innate variability of PD-L1 expression in the blood. Normal healthy PAXgene blood samples showed a range of approximately 1600 copies PD-L1/μL of extracted RNA to 4100 copies PD-L1/μL of extracted RNA (~2.5 fold difference) which should be considered when designing future assays. However, variation from sample to sample should not be surprising and we would expect cancer samples especially from "hot" tumor types (for example Non-Small Cell Lung Cancer and Urothelial Cancer) to present with even higher PD-L1 expression in the blood. Furthermore, measuring the change in PD-L1 expression over time (or course of treatment) may be more informative than a single snapshot, but such theories need to be investigated further.

Expression of PD-L1 in the tissue has widely been used as a biomarker in various cancer indications. However, tissue-based biomarkers present with certain challenges such as limited

material, invasive procedures for procurement, and bias based on heterogeneity of the tumor. Liquid biopsy-based assays can overcome many of the challenges plaguing tissue-based testing. We demonstrated that PD-L1 expression can be detected in PAXgene blood samples in healthy controls. Measuring expression of PD-L1 in a liquid biopsy matrix would open doors for further investigation using PD-L1 as a blood-based biomarker such as longitudinal monitoring of cancer patients during treatment. However, the clinical utility of longitudinal monitoring of PD-L1 has not yet been investigated. We envision blood-based biomarkers tests to work in support of tissue-based assays and contribute to a more accurate overall picture of the disease state.

Fit-for-purpose development and assay transfer efforts were conducted to compare inter-operator variability at external service providers. Importantly, we were not only successful in transferring the assay to a GCP/CLIA/CAP compliant CRO (contract research organization), but also utilized the assay in supporting our clinical translational biomarker study, which highlighted the assay robustness as well as outstanding assay performance.

## 4. Conclusions

Liquid biopsy-based assays hold great potential to build upon the data provided by current tissue biopsy assays. While biomarker testing using tissue biopsies are limited based on tumor site location, material availability, and may only show one small portion of the tumor and/or microenvironment, liquid biopsy testing is able to overcome many of these limitations. Also, liquid biopsy testing can be conducted longitudinally, allowing for close monitoring of disease progression, treatment response, and possible regression with minimal invasiveness.

In conclusion, we developed a liquid biopsy-based ddPCR PD-L1 assay with clinical potential as a pharmacodynamic biomarker assay that can offer complementary information to any available IHC results. While this purely quantitative ddPCR PD-L1 RNA assay has exhibited solid performance, additional development and validation efforts are deem appropriate to bring to clinical testing standards.

## 5. Methods

### 5.1 Sample acquisition

PAXgene RNA tube blood samples from healthy volunteers (n = 35) were procured from Zen Bio (Research Triangle Park, North Carolina). A subset of 5 of these samples were used to assay characterization while the remainder were tested during assay validation at the external service provider.

### 5.2 Ethics approval and consent to participate

The authors state that written informed consent was received from all subjects used in this study and all samples were collected under IRB approved protocols, as defined by the specimen provider ZenBio. Ethical approval was obtained from the ZenBio ethics committee for the collection of specimen samples. After collection, all samples were de-identified by the specimen provider so that no identifying information was available to researchers.

### 5.3 Cell culture

A549 cells (catalog #CCL-185, ATCC, Manassas VA) were grown in F-12K media (catalog #30–2004, ATCC, Manassas VA) plus 10% fetal bovine serum (catalog #35-015-CV, Corning, Amsterdam Netherlands). Cell cultures were maintained at a density below the recommendation by ATCC of 6 x $10^4$ cells/cm$^2$. At passage number three, cells were either treated with IFN

γ (catalog #PHC-4031, ThermoFisher Scientific, Waltham, MA) at 100 ng/mL for 48 hours or left untreated. After 48 hours the cells were counted, harvested, and RNA extracted.

## 5.4 RNA extraction and quality check

RNA was extracted from either 2.5 mL PAXgene blood samples using Qiagen PAXgene miRNA Kit (catalog #763134, Qiagen, Valencia CA) following the manufacturer's instruction or from approximately $3 \times 10^6$ cells using the Qiagen RNEasy Mini Kit (catalog #74107, Qiagen, Valencia CA) following the manufacturer's instructions. RNA quality was assessed by Agilent Bioanalyzer 6000 RNA Nano Kit (catalog #5067–1511, Agilent Technologies, Santa Clara CA) and Nanodrop instrument or Qubit RNA BR Assay Kit (catalog #Q10210, Thermo-Fisher Scientific, Waltham MA) following manufacturer's instruction.

## 5.5 Reverse transcription (RT) of RNA into cDNA

A total of 500 ng RNA was reverse transcribed using ThermoFisher Superscript IV Vilo Kit (catalog #11756500, ThermoFisher Scientific, Waltham MA) following manufacturer's instruction on an Applied Biosystems VeritiDx Thermal Cycler. Each RT reaction was based on a total of 500ng RNA in a volume of 20μL. Multiple reactions for the same sample were combined in one tube.

## 5.6 Standard curve generation

Using ThermoFisher GeneArt synthesis tool, an 850 base pair (bp) synthetic cDNA fragment was designed to span the sections of the PD-L1 gene covered by the selected three TaqMan® PD-L1 assays (see supporting material for more details). The gene fragment was diluted in nuclease free water to an estimated 250,000 copies/μL. This dilution was then tested in droplet digital PCR (ddPCR) and adjusted accordingly. The gene fragment dilution was used to generate a 7-point standard curve from approximately 250,000 copies/μL to 0.25 copies/μL and verified in ddPCR. This standard curve was tested across each primer/probe kit in triplicate on three different days to accurately determine the assay linearity and efficiency in qPCR.

## 5.7 Reference gene selection

Four reference genes, Glucuronidase β, Ribosomal Protein Lateral Stalk Subunit P0, β-2 microglobulin, and TATA Box Binding Protein, (GUSB, RPLP0, B2M, and TBP, respectively) were selected for their presence and stability in PAXgene blood based on previously published data [7,8]. These genes were tested in qPCR to evaluate the stability of the expression in our samples and control material.

## 5.8 Real time PCR

Three commercially available PD-L1 TaqMan® assays, spanning different regions of the PD-L1 gene (Fig 7) were tested for linearity using a PD-L1 cDNA construct. Assays were selected for testing and comparison based on the following criteria:

1. Hs00204257_m1: Best Coverage; amplicon length of 77bp.

2. Hs01125296_m1: Tail end of gene; amplicon length of 87bp.

3. Hs01125301_m1: Most citations (21); amplicon length of 89bp.

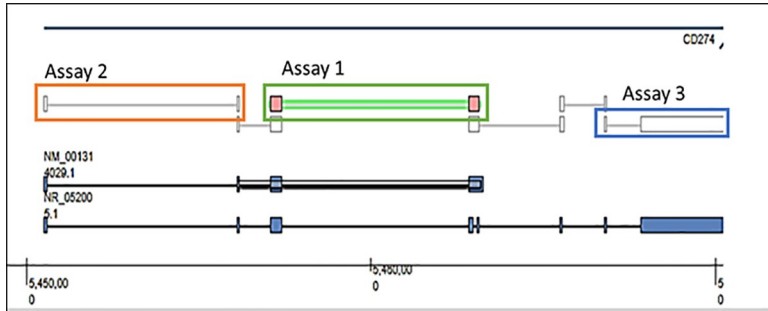

**Fig 7. Schematic showing the PD-L1 gene, highlighting regions which the three PD-L1 TaqMan® assays span.**

Real Time PCR reaction mixes were prepared consisting of 10μL TaqMan® Gene Expression Master Mix (catalog #4369016, ThermoFisher Scientific, Waltham MA), 1μL TaqMan® Primer/Probe set (Table 3), 5μL nuclease free water, and 4μL cDNA (100ng RNA equivalent). qPCR cycling was conducted on the Applied Biosystems ABI7500Dx platform using the following protocol: 50 $^{0}$C for 2 minutes, 95˚C for 10 minutes, followed by 40 cycles of 95˚C for 15 seconds and 60˚C for 1 minute.

## 5.9 Droplet digital PCR (ddPCR)

ddPCR reactions mixes consisted of 11μL of Bio-Rad ddPCR Supermix for probes (No dUTP) (cat #1863023, BioRad, Hercules CA), 1μL TaqMan® Primer/Probe set (Table 3), 2μL cDNA (50ng RNA equivalent), and 8μL H2O. Droplets were generated in the Bio-Rad QX200 droplet generator where the reaction mixtures and oil are automatically combined. Approximately 40μL of the droplet/oil mixture (12,000 to 20,000 droplets) were automatically transferred to a semi-skirted 96-well plate. The plate was sealed with a pierce-able foil heat seal using a Bio-Rad PX1 PCR plate sealer. The sample plate was then loaded and processed on an Applied Biosystems VeritiDx thermal cycler following standard manufacturer's ddPCR amplification conditions. After thermal cycling, the plate was transfer analyzed in a Bio-Rad QX200 plate reader.

## 6. Analysis

### 6.1 "Absolute" quantification in qPCR and ddPCR

To determine "absolute quantification" in qPCR, a standard curve was generated and the equation for the line of best fit was used to extrapolate the copy number of unknown samples using their Cq (cycle quantification value). This standard curve was used to determine the linearity

**Table 3. Primer/probe sets used in assay development.**

| Gene Target | ThermoFisher Catalog Number | Referred to as |
|---|---|---|
| PD-L1 | Hs00204257_m1 | PD-L1 Assay 1 |
| PD-L1 | Hs01125296_m1 | PD-L1 Assay 2 |
| PD-L1 | Hs01125301_m1 | PD-L1 Assay 3 |
| B2M | Hs99999907_m1 | |
| GUSB | Hs99999908_m1 | |
| RPLP0 | Hs99999902_m1 | |
| TBP | HS99999910_m1 | |

and efficiency of each TaqMan® assay. ddPCR is a purely quantitative assay and, in this regard, we were able to use the raw copy numbers for each unknown sample without the use of any standards. GUSB was used as a reference to monitor assay performance in each sample.

## 6.2 Relative quantitation in qPCR and ddPCR

To determine relative quantitation in qPCR we utilized the delta deltaCq (ΔΔCq) method. In this instance, GUSB was used as the reference gene and our control cell line A549 untreated was our reference sample.

ΔCq (unknown sample) = Cq(PD-L1)–Cq(GUSB)

ΔCq (A549 untreated) = Cq(PD-L1)–Cq(GUSB)

ΔΔCq = ΔCq(unknown sample)–ΔCq(A549 untreated)

Relative Expression = 2^-(ΔΔCq)

Relative Quantitation using ddPCR was calculated by first normalizing all samples to GUSB (PD-L1 copies/GUSB copies) and then normalized to A549 untreated cDNA (normalized PD-L1 copies of sample X/normalized PD-L1 copies of A549 untreated cDNA).

## 6.3 Statistical analysis

For the assessment and visualization of data, the statistical software JMP (Version 14.2.0, SAS Institute Inc., Cary NC) was used. A measurement was considered statistically significant when $P<0.05$. Copy numbers were log10-transformed when used as variables in statistical models.

For the assessment of linearity, a linear regression was used, modeling the Cq value by an intercept and the log10-transformed copy number, as this model was judged to be appropriate to describe the data. Extended models that included a quadratic effect showed no significance in these parameters. To compare the three assays jointly, a joint regression (ANCOVA) was fitted using an intercept, the log10-transformed copy number, a categorical variable for assay as well as an interaction term of assay and copy number. JMP's compare slopes procedure calculates the estimates of the slopes compared to overall average slope as well as simultaneous 95% confidence intervals (adjusted for the alternative hypothesis that at least one slope differs from average).

## Supporting information

**S1 Fig. Compare slope graph.** The slopes of each line of best fit from the 7-point cDNA input titration using the compare slopes function in JMP. The average slope (green dot) for each assay is shown along with the upper and lower limits. While this analysis showed no significant difference in the slopes, Assay 3 showed slightly wider upper and lower limits.
(TIF)

**S1 Table. Average Cq value for reference genes.** Average Cq values for four Reference genes, B2M, GUSB, TBP, and RPLP0 in A549 cells untreated and treated with IFN-γ in qPCR.
(DOCX)

**S2 Table. Comparison of three PD-L1 primer/probe assays.** qPCR CT values for PD-L1 Taqman Assays across 7 cDNA input amounts. Cq values from qPCR assays are displayed.
(DOCX)

**S3 Table. Compare slopes analysis for PD-L1 assays.** The compare slopes method resulted in no difference in the slopes of the equation of the line of best fit for our cDNA standard curve.

The average, and upper and lower limits are described.
(DOCX)

**S4 Table.** A. Relative quantitation with qPCR. Relative Quantitation using qPCR is measured using the 2^(-ΔΔCq) method where GUSB is used as the reference gene and A549 untreated cDNA is the reference sample. B. Relative quantitation with ddPCR. Relative Quantitation using ddPCR was calculated by first normalizing all samples to GUSB (PD-L1 copies/GUSB copies) and then normalized to A549 untreated cDNA (normalized PD-L1 copies of sample X/ normalized PD-L1 copies of A549 untreated cDNA).
(DOCX)

**S5 Table.** A. "Absolute" quantification for PD-L1 inn qPCR. "Absolute" quantification for PD-L1 in qPCR was determined by calculating the copy numbers based on a standard curve of the synthetic cDNA construct. This standard curve was verified in ddPCR to ensure accurate copy numbers. B. Absolute quantification for PD-L1 in ddPCR.
(DOCX)

**S6 Table.** A. Inter-assay reproducibility. Inter-assay reproducibility was calculated for ddPCR (absolute quantification), and qPCR (both "absolute" quantification and relative quantification". In each instance average values were obtained for three samples across three different runs. These values were then averaged and %CV was calculated. B. Intra-assay reproducibility. Intra-assay reproducibility was calculated based on three different runs each with five replicates. The percent CV in each run was calculated (displayed below). The average CV from each run was then averaged to get an overall intra-assay CV.
(DOCX)

**S7 Table. Assay transfer data from service provider comparison.** Eight samples were sent to an external service provider to confirm a successful assay transfer. ddPCR results for the service provider and internal data is listed below.
(DOCX)

**S8 Table. Assay validation data from service provider.** Following assay transfer to the service provider, assay validation was performed to calculate LOB, LOD, inter-assay, intra-assay, and inter-operator reproducibility. Where N/A is listed, these samples were not tested in that scenario.
(DOCX)

**S1 File.**
(DOCX)

## Acknowledgments

The authors would like to acknowledge Angela Manginelli and Huilin Xiong for their biostatistical analysis and guidance and Janet Wang for her clinical expertise and guidance.

## Author Contributions

**Conceptualization:** Dennis O'Rourke, Danyi Wang, Ti Cai, Juergen Scheuenpflug, Zheng Feng.

**Data curation:** Dennis O'Rourke, Danyi Wang, Jorge F. Sanchez-Garcia, Waldemar Miller, Zheng Feng.

**Formal analysis:** Dennis O'Rourke, Danyi Wang, Jorge F. Sanchez-Garcia, Waldemar Miller.

**Funding acquisition:** Ti Cai, Juergen Scheuenpflug, Zheng Feng.

**Investigation:** Danyi Wang.

**Methodology:** Dennis O'Rourke, Danyi Wang, Jorge F. Sanchez-Garcia, Maria Perella Cusano, Ti Cai, Zheng Feng.

**Resources:** Dennis O'Rourke, Danyi Wang, Jorge F. Sanchez-Garcia, Maria Perella Cusano, Juergen Scheuenpflug, Zheng Feng.

**Supervision:** Danyi Wang, Zheng Feng.

**Validation:** Dennis O'Rourke, Danyi Wang, Jorge F. Sanchez-Garcia, Zheng Feng.

**Visualization:** Waldemar Miller.

**Writing – original draft:** Dennis O'Rourke, Danyi Wang, Jorge F. Sanchez-Garcia, Maria Perella Cusano, Zheng Feng.

**Writing – review & editing:** Dennis O'Rourke, Danyi Wang, Jorge F. Sanchez-Garcia, Maria Perella Cusano, Waldemar Miller, Ti Cai, Juergen Scheuenpflug, Zheng Feng.

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
