## [Decision Letter · Decision Letter 0]

29 Mar 2021

PONE-D-21-01581

Fit-for-purpose quantitative liquid biopsy based droplet digital PCR assay development for detection of programmed cell death ligand-1 (PD-L1) RNA expression in PAXgene blood samples

PLOS ONE

Dear Dr. O'Rourke,

Thank you for submitting your manuscript to PLOS ONE. After careful consideration, we feel that it has merit but does not fully meet PLOS ONE’s publication criteria as it currently stands. Therefore, we invite you to submit a revised version of the manuscript that addresses the points raised during the review process.

ACADEMIC EDITOR: The results described are interesting and the ms deserves to be published pending minor revision. Please see my comments reported below.

We look forward to receiving your revised manuscript.

Kind regards,

Maria Stefania Latrofa

Academic Editor

PLOS ONE

Journal Requirements:

"This study was funded by EMD Serono Inc., a part of Merck KGaA Darmstadt,

Germany. EMD Serono/ Merck KGaA is the employer of all authors (with the exception

of Maria Cusano, who was previously employed by EMD Serono) at the time of

submission."

We note that one or more of the authors are employed by a commercial company: "Merck KGaA,

Additional Editor Comments:

Line 276: specify how many blood samples and /or healthy volunteers were used/enrolled

Lines 277-281: remove being the same sentences reported in the paragraph 6.2

Lines 289-303: rearrange the paragraphs 6.3, 6.5 and 6.6 into one

Lines 347-355: rearrange the paragraphs 7.1 and 7.2 into one

Lines 84, 124, 163-165 and 356: Specify why the relative quantification has not been also described for ddPCR. Indeed, a Supplementary Table 4B has been included

Line 131: check the value (0.44), in suppl. Table 4A the value for assay 2 is 0.43.

Line 145: describe the Average PAX gene Absolution Expression obtained in qPCR for all three assays used. I suggest including in the text the supplementary Table 5A and 5B

Line 146: specify why only five blood samples were tested in ddPCR, being 35 healthy volunteers were enrolled (as reported in line 29)

Lines 163-164 and lines 168-169: I would suggest avoiding to write the same sentence several times; for example rephrase as follows “Inter- and intra- assay reproducibility…..primer/probe kits”

Table 1 and Table 2: add the average value described in the text in both Table, to make results clearer

Line 177: Add (Table 2) after “comparable”

Lines 232-238: in the results section, it is not clearly described which is the best among the three PD-L1 assays used. Indeed, in the discussion is reported that the “ultimately PD-L1 Assay 2 was chosen for assay validation with the external service provider based on the lowest inter-assay CV and intra-assay CV. In ddPCR, Assay 2 showed similar median and mean expression levels to Assay 1 but with slightly lower range of expression whereas Assay 3 had lower median and mean values.” So, it is not clear to me which is the best assay both in the qPCR and in the ddPCR. In the result section (lines 163-179) briefly describe the inter- and intra-assay CV values obtained in qPCR and ddPCR, for each assay used.

Indicates the supplementary Tables 6A, B, 7 and 8 in the text.

Line 240-243: delete the sentences, starting with “The ddPCR offers absolute….”

Discussion needs to be improved emphasizing more in depth the main results obtained for ddPCR and on the best Taq-Man assay identified for this tool.

Reviewers' comments:

Reviewer's Responses to Questions

**Comments to the Author**

1. Is the manuscript technically sound, and do the data support the conclusions?

Reviewer #1: Yes

2. Has the statistical analysis been performed appropriately and rigorously? 

Reviewer #1: Yes

3. Have the authors made all data underlying the findings in their manuscript fully available?

Reviewer #1: Yes

4. Is the manuscript presented in an intelligible fashion and written in standard English?

Reviewer #1: Yes

5. Review Comments to the Author

Reviewer #1: The manuscript regarding the development of a fit-for-purpose quantitative liquid biopsy based droplet digital PCR

for detection of programmed cell death ligand-1 (PD-L1) RNA expression in PAXgene blood samples is well-written in good level of English language. The quantitative ddPCR assay for the detection of PD-L1 mRNA expression developed and validated using PAXgene RNA blood samples demonstrated promising linearity, reproducibility, limit of blank and limit of detection compared to quantitative PCR with Taqman assay. The developed ultrasensitive assay could be useful in clinical applications with a potential in screening,longitudinal monitoring and disease progression detection.

I therefore suggest to publish the manuscript in the Journal.

The quality level of figures could be improved maybe with higher dot per inch output of all the figures.

6. PLOS authors have the option to publish the peer review history of their article (what does this mean?). If published, this will include your full peer review and any attached files.

Reviewer #1: No

---

## [Author Response · Author response to Decision Letter 0]

9 Apr 2021

Dear Reviewer and Editors,

Thank you for the kind comments and edit suggestions. We have listened to each comment and made the appropriate changes in the manuscript which we know believe is suitable for publication. Detailed explanation about our edits can be found in the "Response to Reviewers" attachment. If any further edits or changes are needed please reach out again and we are happy to accomodate.

Thank you on behalf of all authors,

Dennis O'Rourke

---

## [Editor Report · Decision Letter 1]

15 Apr 2021

Fit-for-purpose quantitative liquid biopsy based droplet digital PCR assay development for detection of programmed cell death ligand-1 (PD-L1) RNA expression in PAXgene blood samples

PONE-D-21-01581R1

Dear Dr. O’Rourke,

We’re pleased to inform you that your manuscript has been judged scientifically suitable for publication and will be formally accepted for publication once it meets all outstanding technical requirements.

Kind regards,

Maria Stefania Latrofa

Academic Editor

PLOS ONE

Additional Editor Comments:

In my opinion the revised version of this paper is now acceptable for publication. See below just some suggestions

Lines 247-250: I suggest deleting the sentence “Initially it was our intention to develop a RT-qPCR assay due to …. prompting the switch to” and starting the sentence with “The ddPCR offers absolute…”

Lines 244-245:I suggest deleting the sentence "Such experiments are planned, testing our ddPCR assay with matched cancer tissue samples to examine correlations between blood and tissue PD-L1 expression."

---

## [Editor Report · Acceptance letter]

28 Apr 2021

PONE-D-21-01581R1 

Fit-for-purpose quantitative liquid biopsy based droplet digital PCR assay development for detection of programmed cell death ligand-1 (PD-L1) RNA expression in PAXgene blood samples 

Dear Dr. Feng:

I'm pleased to inform you that your manuscript has been deemed suitable for publication in PLOS ONE. Congratulations! Your manuscript is now with our production department. 

Kind regards, 

on behalf of

Dr. Maria Stefania Latrofa 

Academic Editor

PLOS ONE